# Optical Hydrogen Sensing Properties of e-Beam WO_3_ Films Decorated with Gold Nanoparticles

**DOI:** 10.3390/s23041936

**Published:** 2023-02-09

**Authors:** Elena Colusso, Michele Rigon, Alain Jody Corso, Maria Guglielmina Pelizzo, Alessandro Martucci

**Affiliations:** 1Department of Industrial Engineering, University of Padova & INSTM, Via Marzolo 9, 35131 Padova, Italy; 2CNR—Istituto di Fotonica e Nanotecnologie, Via Trasea 7, 35131 Padova, Italy; 3Department of Information Engineering, University of Padova, Via Gradenigo 6, 35131 Padova, Italy

**Keywords:** hydrogen sensing, tungsten oxide, e-beam deposition, gold nanoparticles, optical sensors

## Abstract

Tungsten oxide thin films with different thicknesses, crystallinity and morphology were synthesized by e-beam deposition followed by thermal treatment and acid boiling. The films with different surface morphologies were coated with gold nanoparticles and tested as optical sensing materials towards hydrogen. X-ray diffraction, scanning electron microscopy, ellipsometry and UV-VIS spectroscopy were employed to characterize the structural, morphological and optical properties of the film. We demonstrated a good response towards hydrogen in air, reaching a good selectivity among other common reducing gases, such as ammonia and carbon monoxide. The sensitivity has been proven to be highly dependent on the thickness and crystallinity of the samples.

## 1. Introduction

Tungsten oxide (WO_3_) is a transitional metal oxide that has attracted great interest within the researchers’ community in recent years. Its large band gap (between 2.60 to 3.25 eV [1]) coupled with high chemical stability, no toxicity and interesting physicochemical proprieties made this oxide a perfect candidate for a wide range of applications, e.g., electrochromic material for smart windows [2], contact or injunction material for solar cells [3], photoelectrochemical water splitting [4], photocatalyst [5] and sensing material for chemical gas detection [6,7,8,9].

A high number of optimized methods for sensing nitrogen dioxide, ammonia, hydrogen sulfide, hydrogen and ethanol are reported in the literature [6,9,10,11]. Nanostructuring plays a pivotal role in the properties of the sensor, since structures such as nanorods, nanowires, nanoplates, etc., offer a dramatically higher surface-to-volume ratio in comparison with the bulk, enhancing the sensing performances [9]. Thus, different synthesis and deposition techniques were developed to improve the sensitivity and selectivity of the material. We can name, for instance, the sputtering techniques [12,13], thermal evaporation [14], pulsed laser deposition [15], electrostatic spray deposition [16], sol-gel process [17] and e-beam deposition [10]. The e-beam technique offers the advantage of high reproducibility, high thermal efficiency and extensive control of film structure.

Among all the different analytes, tungsten oxide shows very good performances in hydrogen sensing, becoming an established conductometric and chemoresistive sensor [6,9,14,18,19,20]. On the contrary, only a few papers characterize its behavior as an optical gas sensor, mainly using platinum and palladium as hydrogen activators, enhancing hydrogen reactivity and its spillover onto the metal oxide’s surface [12,13,15,16,21]. Indeed, an attractive property of tungsten oxide when it is exposed to hydrogen is its gaschromism: the color turns from transparent to blue due to a reduction of the tungsten ions from 6^+^ to 5^+^ [22]. 

One of the first approaches to avoid the usage of such expensive metals is reported by the Wlodarski research group [23], which used a wet chemistry method to precisely deposit gold nanoclusters onto RF-sputtered WO_3_ thin films. The so-obtained Au/WO_3_ sensors exhibited gasochromic properties when exposed to hydrogen in the 800–1000 nm range. Recently, we have demonstrated that gold can provide hydrogen sensing activity of Au/WO_3_ films with a specific mechanism occurring at the Au-WO3 interface, strongly dependent on the nanoparticles’ dimension [17]. 

In this paper, we investigated the optical sensing performances of WO_3_ thin films deposited over fused silica glass by e-beam evaporation and decorated with gold nanoparticles synthesized by wet chemistry. We evaluated the effect of thickness, crystalline structure and morphology on the optical gas sensing response. Flat thin films were compared with nanostructured films with platelet surface obtained by acid oxidation [24]. We characterized the samples using glazing X-ray diffraction (XRD), ellipsometry, UV-VIS spectroscopy and scanning electron microscopy (SEM). Compared to most of the works reported in the literature, where the sensors were tested towards hydrogen balanced in inert gases, we simulated a more realistic operative environment by monitoring the variation in absorbance during the exposition to 1% hydrogen in air.

## 2. Materials and Methods

Tungsten oxide amorphous thin films were deposited onto a fused silica glass substrate by e-beam evaporation [25]. The depositions were performed using an IONVAC^®^ e-Beam & Joule Evaporation system, using tungsten (99.9% in purity in 3–6 mm pellet form) as evaporation material. The distance between the target and the bare substrates was set to 40 cm. The sample holder was heated at 150 °C and spun during the entire process to reach homogeneity in the coatings’ thickness. Residual pressure inside the chamber of around 1.5 × 10^−5^ mbar provided the oxygen to oxidize the pure metal before the deposition over the substrate. The depositions were run with a rate of 10 nm/min and a current of emission of 28 mA in order to produce samples of around 500 nm and 300 nm in thickness. The deposited material was monitored during the process by an Inficon film thickness monitor. After the deposition, the samples were heat-treated at 550 °C in air for two hours (heating ramp 5 °C/min) to promote crystallization.

Tungsten oxide nanoplate thin films were formed by immersing an amorphous WO_3_ layer (300 nm thickness deposited on fused silica by the e-beam process) in nitric acid solution (1.5 M HNO_3_ in water) at 50 °C for three hours [24]. At the end of the process, the samples were washed with fresh deionized water and dried over a hot plate at 150 °C for 30 min. Finally, the films were thermally annealed at 550 °C for two hours (heating ramp 5°C/min) to remove the intraplane water and promote crystallization.

Mercaptoundecanoic acid (MUA)-capped Au NPs (average diameter of 13 nm) were synthesized with a citrate synthesis method [26]: a solution of sodium citrate dihydrate (40 mM in MilliQ water) preheated at 70 °C was quickly added to a 0.5 mM tetrachloroauric acid (Sigma-Aldrich) water boiling solution. After the solution turned red-wine-colored, it was stirred for an additional 15 min and then cooled down to room temperature. A solution composed of 4.5 mg of 11-mercaptoundecanoic acid (Sigma-Aldrich, Burlington, MA, United States), 10 mL of water and 0.3 mL ammonia (Sigma-Aldrich, 25% in water) was slowly added under constant stirring and then left to stir for 2 h. Then, the NPs were clustered by 1 M hydrochloric acid and decanted overnight. The supernatant was finally discarded and the sediment was dispersed in water to have a final concentration of 80 mM.

The deposition of Au nanoparticles (NPs) onto the WO_3_ thin films was performed using the spin coating technique (3000 rpm, 1 min, two times) by diluting the starting AuNPs solution with methanol (40 mM). The surface of WO_3_ films was previously functionalized with an aminosilane layer by immersing the sample in a solution of 3-aminopropyltrimethoxysilane (APTMS, Sigma-Aldrich) in anhydrous toluene (4% *v*/*v*) at 70 °C for 10 min, followed by rinsing with toluene and N_2_ blow-drying [27]. The films were stabilized at 100 °C for 10 min on a hot plate after the first deposition and finally dried at 300 °C for a further 10 min. In the case of the nanostructured thin films, gold nanoparticles were directly spun onto the nanostructured layers using the same recipe.

The crystalline compositions of the samples were determined by X-ray diffraction (XRD, Philips PW1710 diffractometer with Cu Kα Ni filtered radiation, equipped with grazing incidence X-ray optics). The analysis was performed with an incident angle of 1° in the 20°–80° 2θ range. The morphologies of the as-obtained films were observed on a scanning electron microscope (FE-SEM, Zeiss Sigma HD, 5 kV, InLens detector, Carl Zeiss S.p.A, Milano, IT). Ellipsometry spectroscopy was employed to determine the thickness and refractive index of the films (V-VASE J.A. Woollam). Measurements were conducted in the 400–1200 nm wavelength range at three incident angles (55°–65°–75°). Since the WO_3_ is a nonabsorbing material in the chosen spectral range [28], the measured ellipsometric parameters were fitted with a Cauchy model using the WVASE J.A. Woollam dedicated software. Optical transmittance and reflectance of the samples were collected using a Jasco V-650 spectrophotometer equipped with an integrating sphere in the 200–900 nm wavelength spectral range. The optical band gap of the samples was calculated by using the Tauch method adapted for the thin film, according to the following equation [29]:(1)$$\alpha =-\left(\frac{1}{d}\right)\;\ln\left(\frac{T}{100}\right)\;\;\;,$$
where *d* is the thickness of the coating measured by ellipsometry and *T* is the transmittance. An indirect allowed transition was assumed [30,31]; thus, the values were calculated from the plot α·hν12 versus energy in eV.

The optical gas sensing properties of the samples were tested using a customized Harrick flow cell (Harrick, Pleasantville, NY, USA) coupled with a Jasco V-650 UV-Vis spectrophotometer, setting the operating temperature (O.T.) at 250 °C and using hydrogen 1% balanced in dry air as the target gas. The gas flow rate was set at 8 mL/min. Cross-sensitivity tests were performed towards carbon monoxide (100 ppm balanced in dry air) and ammonia (1000 ppm balanced in dry air).

## 3. Results and Discussion

Tungsten oxide thin films were deposited on fused silica glass by e-beam evaporation, tuning the conditions to deposit films of about 300 and 500 nm in thickness. The as-deposited samples were amorphous in nature but they cracked within one week due to the lattice mismatch with the substrate that introduces tensions inside the film, compromising the optical properties. To relax the structure and reduce the formation of cracks, the samples underwent a thermal treatment. In order to preserve the amorphous nature, the temperature of the annealing was set to 300 °C. Alternatively, samples were treated at 550 °C to induce crystallization as already reported in the literature [13]. Finally, the surface of some pristine samples was modified by treating the film in warm nitric acid, which leads to the formation of hydrate oxides, with a layered structure [24,32]. After the acid treatment, the films were treated at 550 °C to remove all the intraplane water and crystallize the oxide. From now on, we will refer to the different samples according to the acronyms reported in Table 1.

The crystalline phase of the pristine (without gold) tungsten oxide thin films was characterized by X-ray diffraction (XRD). Figure 1 shows no diffraction peaks in the coating after the stabilization at 300 °C (A-WO_3_). On the contrary, the expected phase transition from amorphous to crystalline occurs during the thermal annealing at 550 °C. The observed diffraction peaks belong to the monocline WO_3_ system (ICDD card no. 83-0951, space group P21/n). The most dominant reflections are at 24.38° coming from the (200) plane and at 34.20° due to the (202) plane of the monoclinic structure. The pattern of the nanostructured film (Nano-WO_3_) shows peaks that belong only to tungsten trioxide, confirming the absence of any other phases, such as hydrated oxides.

The thickness of the amorphous layer was measured to be 470 nm, showing a small variation in comparison to the one estimated directly during the deposition process (set to 500 nm). The refractive index was 2.06 at 650 nm, in agreement with the literature [17]. The optical band gap calculated by using the Tauc and Menth method was 3.26 eV, matching the results reported by previous works for amorphous WO_3_ [1,31]. After annealing at 550 °C, the sample C-WO_3__500 showed an average thickness of 420 nm and a refractive index of 2.20. The slight decrease in thickness and the variation in refractive index are associated with densification and crystal formation. A similar refractive index was measured for the crystallized sample deposit with a lower set thickness, measured to be 340 nm after crystallization (C-WO_3__300). The optical band gap of the films decreases from 3.26 to 2.71 eV after crystallization, according to [30]. Transmittance and reflectance spectra of the crystallized films showed pronounced oscillations due to optical interference. This phenomenon is also visible to the naked eye since a green/blue or pink reflection appears when the thin film is observed directly from above or slightly tilted, respectively (see Appendix A).

In order to perform the gas sensing tests, the WO_3_ samples were optically activated by depositing over their surfaces MUA-capped gold NPs synthesized by the Turkevich method as reported in the experimental section. XRD reveals the presence of diffraction peaks at 38.2° and 44.4° of cubic Au (ICDD card 04-0784) on the gold-covered samples (Appendix A).

Figure 2a shows the morphology of a pristine crystalline film after annealing at 550 °C, characterized by a nonporous layer with some cracks over the surface, probably associated with residual stresses. A representative image of a gold-covered sample is visible in Figure 2b. The NPs are homogeneously distributed all over the surface, with a few spots showing aggregations. The average dimension is ~23.2 nm (C-WO_3__500) and ~21.9 nm (C-WO_3__300), with a total surface coverage of 36.2% and 35.2%, respectively, as determined by image analysis (see Appendix A). Finally, it is worth noticing the high homogeneity of the Nano-WO_3_ coating, with platelets that homogenously cover the entire film surface without aggregates of nanoplatelets or spots in which the bare substrate is visible (Figure 2d). The platelets have a size of around 400–1000 nm in length and a thickness within the 50–60 nm range. The gold NPs appear well distributed over the nanoplatelets’ surface without the formation of aggregates.

The hydrogen sensing properties of the samples were tested by monitoring the variation in absorption spectra in the presence of the analyte. Pristine WO_3_ films (without gold) did not show any sensing response. Figure 3 shows the spectra of the gold-decorated samples in dry air and in 1% H_2_ (balanced in dry air) at 250 °C O.T. When exposed to hydrogen, all the samples show a small change in absorbance in the 550–700 nm range, corresponding to a blue shift of the localized surface plasmon resonance (LSPR) of Au NPs. The observed peak shift agrees with results reported in the literature on tungsten trioxide films doped with gold nanoparticles [17] or other metal oxide systems [33]. It can be attributed to changes in the dielectric properties of the surrounding medium as a result of electron injection into the n-type oxide matrix and a reduction of tungsten oxide. In addition, it has been reported that the refractive index of WO_3_ decreases in the presence of hydrogen [17].

To better visualize the different sensitivity of the samples, we define the response parameter optical absorbance change (OAC = A_gas_−A_Air_), measured as the difference in absorbance between samples exposed to the hydrogen gas and air. As visible from Figure 4, all the OAC curves are characterized by a minimum in the 500–700 nm region, even if the variation in the OAC is quite different for the samples, with a maximum variation for the C-WO_3__300 sample. The increased sensitivity compared to the C-WO3_500 sample can be explained by considering the reduced film thickness and the nanoparticles’ interface perimeter. As reported in the characterization, the average diameter for the C-WO_3__300 sample is slightly lower (about 22 nm) compared to the C-WO_3__500 (about 24 nm). Since the surface coverage is comparable, we can assume that the total interface perimeter is slightly higher for the C-WO_3__300, thus yielding a higher optical response, in accordance with [17].

In fact, when a thin film of WO_3_ is covered by a noble metal catalyst (such as Pt, Pd or Au), it changes color from transparent to blue when exposed to H_2_ [12,13,15,16,21,22]. This phenomenon can be explained on the basis of a multistep process: When molecular hydrogen interacts with the catalyst, it first adsorbs on the metal surface, weakening the H-H chemical bond and dissociating into two atomic species. These atomic hydrogens are then free to diffuse on the metal surface and, when reaching the border of the metallic nanostructure, migrate on the tungsten oxide. The latter migration step is also known as “spillover”. When the hydrogen atoms reach the oxide surface, they can easily diffuse within the bulk and spontaneously react, reducing the valence state of tungsten atoms from 6^+^ to 5^+^ and forming tungsten bronze H_x_WO_3_, which appears blue under visible light.

Recently, we demonstrated [17] that in the case of Au-activated WO_3_, there is heterolytic adsorption of H_2_ at the Au/WO_3_ interface, and therefore, a larger interface region (i.e., smaller nanoparticles) means a higher H_2_ intake.

Each sample has been dynamically tested at the wavelength corresponding to the maximum variation in the OAC curve, i.e., at 640 nm (for A-WO_3_), 630 nm (for C-WO_3__500), 655 nm (for C-WO_3__300) and 620 nm (for Nano-WO_3_). The time-resolved curves are reported in Figure 5.

A-WO_3_ shows poor sensing dynamics, since the signal never reaches stability both in hydrogen and in air (Figure 5a). This behavior is probably due to the slow kinetics of adsorption and desorption, which do not allow the sensor to reach equilibrium in the new gas environment. Furthermore, the drift of the signal is probably correlated to irreversible reactions on the sensor’s surface since the signal never recovers the original baseline even with a very long time of exposure to air. The crystallization of tungsten oxide remarkably increases its sensing properties (Figure 5b), allowing it to reach an equilibrium in the hydrogen atmosphere and, consequently, stability of the signal. This could be related to a faster diffusion of hydrogen in polycrystalline WO_3_ films compared to the amorphous ones [34].

The decrease in the thickness of the layer (C-WO_3__300) seems to enhance the adsorption reaction with hydrogen, decreasing the response time to 1.8 min (Figure 5c). On the contrary, irreversible adsorption or desorption characterized by very low kinetics still occurs, making the recovery time comparable to the C-WO_3__500. Indeed, during the oxidation process, the desorption of H^+^ and the adsorption of O^2−^ are slow due to the small thermodynamic desorption barrier of H^+^ [35].

The best behavior in terms of response time and reproducibility has been demonstrated by the Nano-WO_3_. Indeed, the dynamic response in Figure 5d shows an adsorption time of only 53 s, about half of the time required by the C-WO_3__300, which has the best performance among the dense films. This faster response could be explained considering the reduced thickness of the nanoplates (50–60 nm) and the increased surface area compared to the flat films, in agreement with [9]. A similar response time (55 s) was reported for optical hydrogen sensing of WO_3_ films (deposited by RF magnetron sputtering) decorated with Pt nanoparticles [13], while it is significantly lower compared to previous results on hydrogen sensing of Au-WO_3_ films obtained by solution processes [17]. Xiang et al. demonstrated a response of 8 s for WO_3_ nanorods decorated by gold nanoparticles in resistive hydrogen sensing at 290 °C [36]. They showed an Au loading dependence effect in the sensor sensitivity, with an optimal amount of 0.5% Au. Compared to other metal oxides/Au optical hydrogen sensors, the response time of WO_3_/Au is close or even lower [37], as summarized in Table 2. A faster response can be achieved in resistive or conductometric sensors by employing more expensive catalysts (Pt or Pd) and operating in an inert atmosphere. 

We tested the cross-sensitivity of WO_3_ towards carbon monoxide (CO) and ammonia (NH_3_). As visible in Figure 6, the samples display chemical inertia to these reducing gases (the observed fluctuations are due to a consequential overshoot and undershoot of the thermal controller), confirming sensing activity only towards hydrogen at the specific wavelength of analysis.

Finally, we characterized the samples after the sensing tests to check the stability of the NPs in the harsh environment. No significant variations were observed from SEM and XRD analysis, confirming the stability in the sensing environment (Appendix A).

## 4. Conclusions

In this work, a tungsten oxide thin film decorated by gold NPs was demonstrated to be a promising material for the optical detection of hydrogen in dry air, without using platinum or palladium, reaching good selectivity among other common reducing gases, such as ammonia and carbon monoxide. The thin films obtained by e-beam deposition have a dense and nonporous structure, which does not allow the target gas to diffuse inside the bulk material, limiting the effective metal oxide involved in the process. We showed that the crystallinity, thickness and morphology play a crucial role in the sensing performance of the film. As a consequence, the maximum sensitivity in terms of the difference in absorbance has been obtained for the thinner crystalline sample. However, the response and recovery times at 250 °C towards 1% H_2_ are in the range of minutes, probably due to irreversible reaction at the sensor’s surface or slow kinetics of interactions. The behavior is dramatically enhanced by nanostructuring the metal oxide, thus yielding a response time that is almost two times faster than that of the dense thin films.

## Figures and Tables

**Figure 1 sensors-23-01936-f001:**
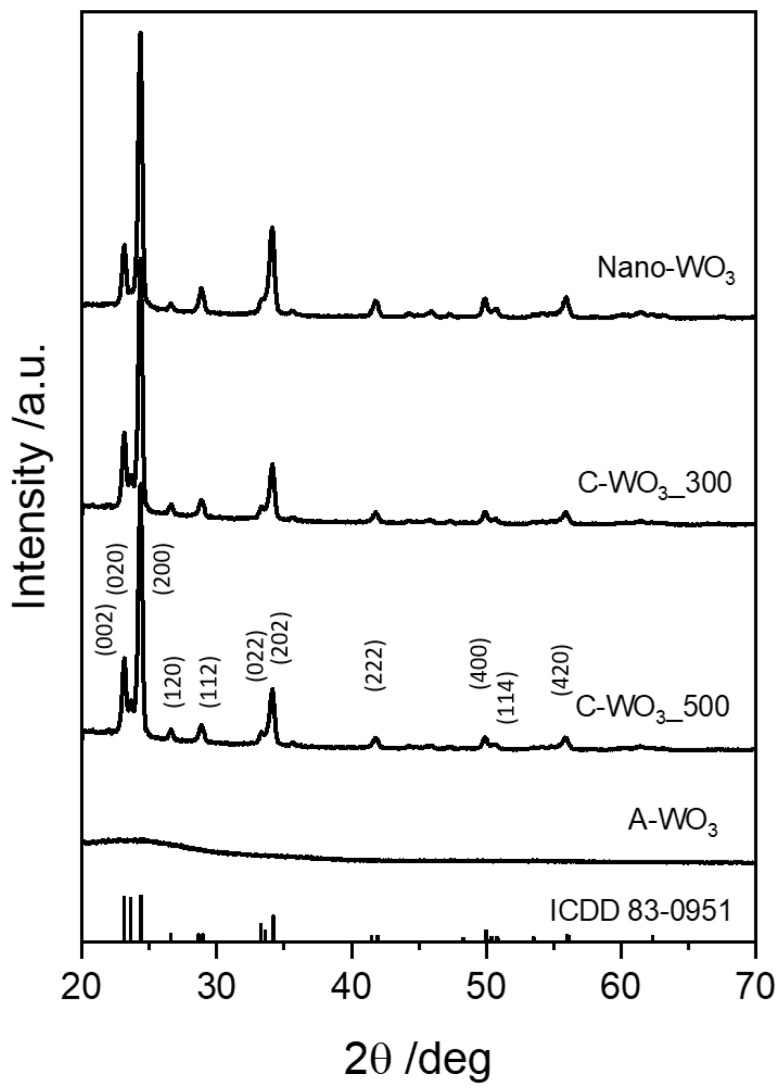
XRD patterns of the e-beam WO_3_ thin films over fused silica substrates.

**Figure 2 sensors-23-01936-f002:**
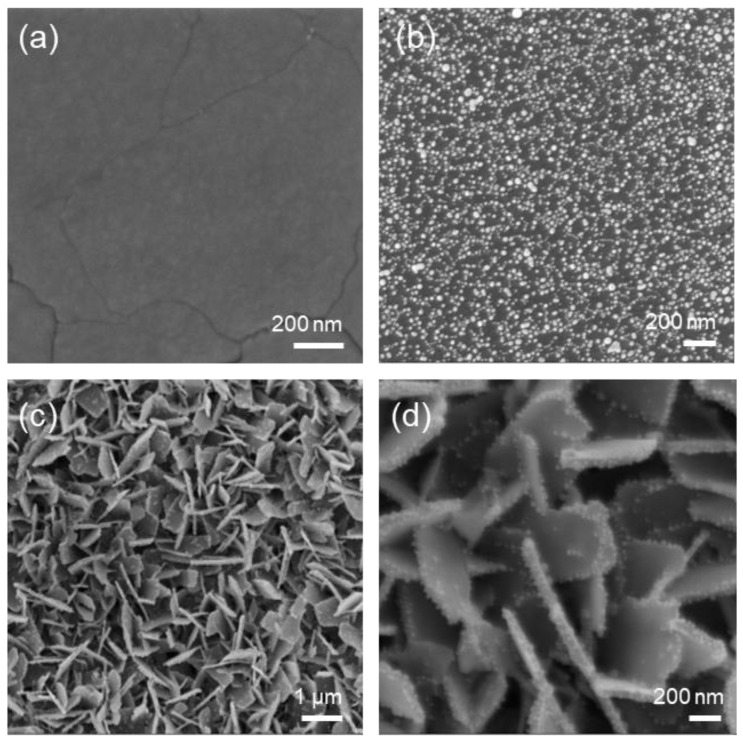
SEM micrographs of the prepared samples: (**a**) pristine crystalline WO_3_; (**b**) gold-covered C-WO_3__500 film; (**c**,**d**) gold-covered Nano-WO_3_ film at different magnifications.

**Figure 3 sensors-23-01936-f003:**
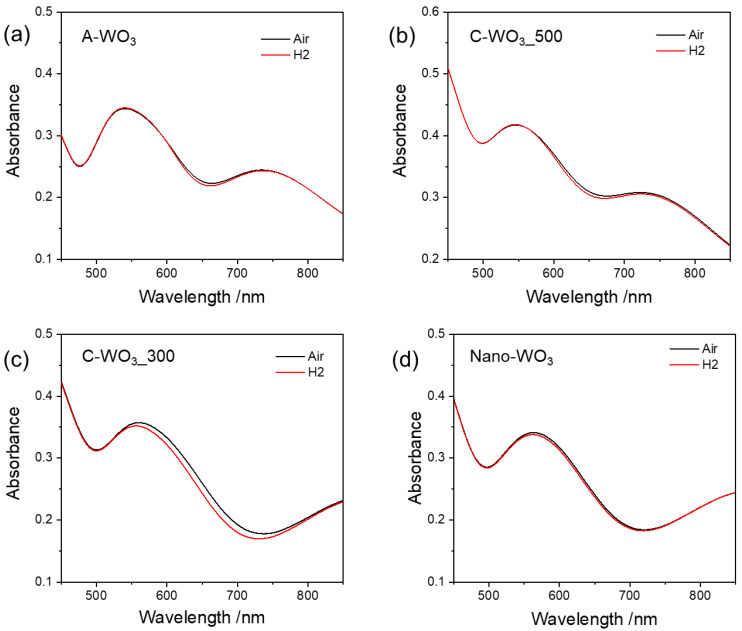
Optical absorption spectra of (**a**) A-WO_3_, (**b**) C-WO_3__500, (**c**) C-WO_3__300 and (**d**) Nano-WO_3_ measured in air (black line) and during exposure to H_2_ (red line) at 250 °C O.T.

**Figure 4 sensors-23-01936-f004:**
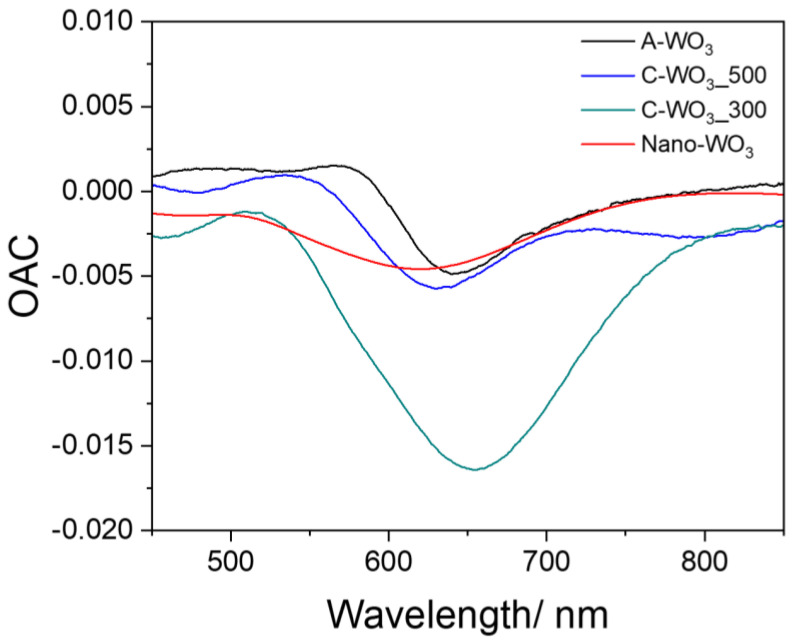
Optical absorption change (OAC) as a function of radiation wavelength for the tested samples.

**Figure 5 sensors-23-01936-f005:**
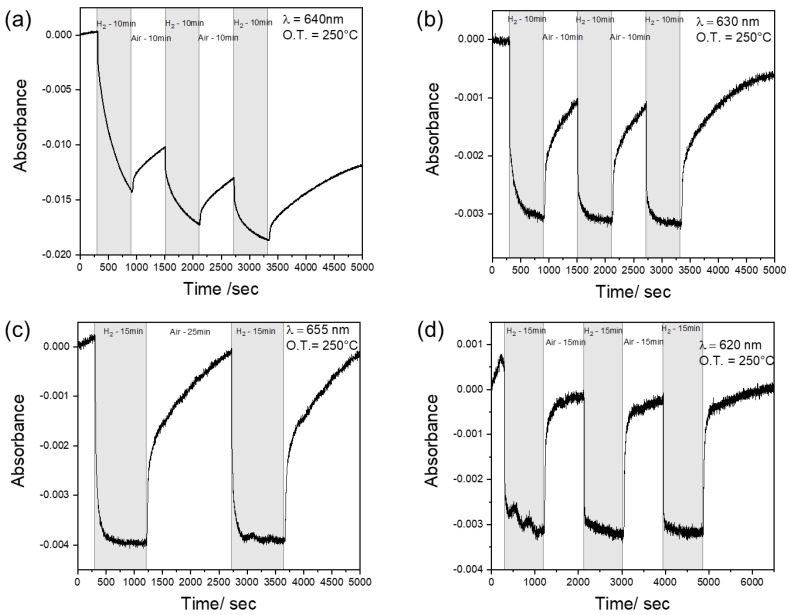
Dynamic response towards hydrogen detection at 250°C for the different tested samples: (**a**) A-WO_3_ at 640 nm; (**b**) C-WO_3__500 at 630 nm; (**c**) C-WO_3__300 at 655 nm; (**d**) Nano-WO_3_ at 620 nm.

**Figure 6 sensors-23-01936-f006:**
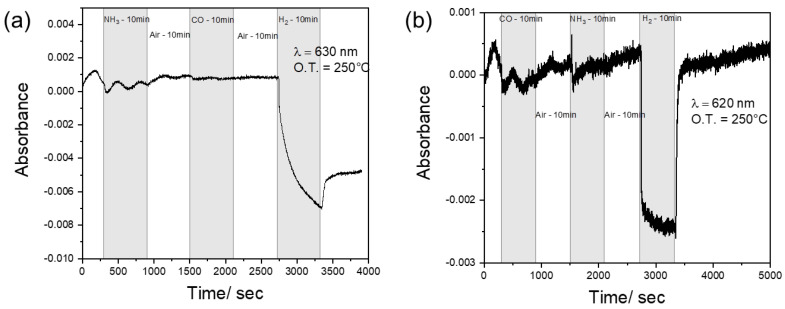
Cross-sensitivity test at 250°C towards H_2_, CO and NH_3_ gases for the C-WO_3__500 (**a**) and Nano-WO_3_ (**b**).

**Table 1 sensors-23-01936-t001:** List of the tested samples with the corresponding thickness and annealing conditions.

Sample	WO_3_ Phase	Processing Conditions	Thickness nm
A-WO_3_	amorphous	E-beam deposition + thermal stabilization @300 °C × 3 h	470
C-WO_3__500	crystalline	E-beam deposition + thermal treatment @550 °C × 2 h	420
C-WO_3__300	crystalline	E-beam deposition + thermal treatment @550 °C × 2 h	340
Nano-WO_3_	crystalline nanoplates	E-beam deposition + HNO3 1.5 M @ 50 °C × 3 h + drying over plate @150 °C × 30 min + thermal treatment @550 °C × 2 h	/

**Table 2 sensors-23-01936-t002:** Comparison of the sensing response in present work with that in previously reported works.

Sample	Process	Operating Temperature	Type of Response	Hydrogen Concentration	Response Time	Reference
C-WO_3__500	E-beam + NPs spinning	250 °C	Absorbance	1% in air	3.1 min	This work
C-WO_3__300	E-beam + NPs spinning	250 °C	Absorbance	1% in air	1.8 min	This work
Nano-WO_3_	E-beam + acid treatment + NPs spinning	250 °C	Absorbance	1% in air	53 s	This work
NiO-Au	Sol-gel	300 °C	Absorbance	1% in air	2.5 min	[37]
ZnO-Au	Sol-gel	300 °C	Absorbance	1% in air	70 s	[37]
TiO_2_-Au	Sol-gel	Room T	Absorbance	1% in air	3 min	[33]
TiO_2_-Pt	Sol-gel	Room T	Absorbance	1% in air	20–40 s	[33]
WO_3_-Au	Sol-gel	200 °C	Absorbance	5% in Ar	19 min	[17]
WO_3_-Pt	Sol-gel	200 °C	Absorbance	5% in Ar	20 s	[17]
WO_3_-Pt	RF-Magnetron Sputtering	300 K	Absorbance	1% in air	55 s	[13]
WO_3_-Pt	RF-Magnetron Sputtering	423 K	Absorbance	1% in air	360 s	[13]
Porous/nanowires WO_3_-Pt films	Solvothermal And PS template + sputtering	Room T	Transmittance	4% in Ar	24.8 s	[12]
WO_3_-Pd	Electrostatic Spray deposition	Room T	Color (reflectance)	1% in N_2_	15–30 s	[16]
WO_3_-Au Nano rods	Hydrothermal	290 °C	Resistance	50 ppm	8 s	[36]
WO_3_-Pt	E-beam	80 °C	Current	0.1% in air	40 s	[38]
WO_3_-Au	E-beam	200 °C	Current	0.1% in air	60 s	[38]
WO_3_-Pt	Sputtering	200 °C	Current	1% in air	200 s	[14]

## Data Availability

The data presented in this study are available on request from the corresponding author.

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
