# Peer review of "Optical Hydrogen Sensing Properties of e-Beam WO3 Films Decorated with Gold Nanoparticles"

_sensors, 2023, doi:10.3390/s23041936_

Round 1

Reviewer 1 Report

Colusso et. al. prepare Au nanoparticle coated WO3 films for the optical sensing of hydrogen. They prepare the WO3 films in a few different methods and find greater response and selectivity for nanostructured films with high crystallinity. The results are interesting, figures are well presented and the manuscript is well written. The paper is suitable for publication in sensors.

Some changes the authors should adress:

There are many instances where subscripts or superscripts are missing.

I cannot find equation (1) within ref 29. The authors should check this section around band gap calculation carefully.

The paper would benefit from a discussion comparing the obtained results to those already in the literature.

Author Response

Please, find attached our point-by-point response.

Reviewer 2 Report

This work demonstrates a tungsten oxide thin film with gold nanoparticles for optical gas sensing. The WO3 thin film deposited by the e-beam has varied crystallinity, surface morphology, and thickness, among which the 300 nm crystalline WO3 showed the best sensitivity in detecting 1% H2 in dry air. While such crystalline thin film has a slow response and recovery time because of the nonreversible reaction at the surface. The adoption of nanostructured WO3 solves this problem as it has a reduced thickness compared to thin films. The cross-sensitivity of the gas sensing at the specific wavelength 630 nm was confirmed by placing the sensor in the environment of H2, co, and NH3 gases.

1.       Since the sensing is in fact provided by the gold nanoparticles instead of WO3, has the author compared the sensing performance to nanoparticles themselves or other oxide decorated with nanoparticles?

2.       The authors attributed the shorter response time of nanostructured WO3 to the reduced thickness of around 50-60 nm compared to the 300 nm thin film. If this comparison is made between 50 nm thin film and nanostructured WO3, the conclusion would be more convincing.

3.       The nanostructured WO3 showed less sensitivity than the 300 nm thin film. Does the author have any explanation for this difference?

4.       The author used glass slides as the substracte for depositing WO3, which leads to cracks because of the residual strain. Are there any better choices for the substrate considering the lattice mismatch? What is the reason to choose fused silica glass as the substrate in this study?

Author Response

(The authors gave the same response as above.)

Reviewer 3 Report

Optical hydrogen sensing properties of e-beam WO3 films decorated with gold nanoparticles

The present manuscript reports the formation of gold nanoparticles deposited on top of tungsten oxide thin films. The aim of the work is the synthesis of a device used for hydrogen sensing. The tungsten oxide thin films are deposited using e- beam deposition technique, and after some thermal treatment, the tungsten oxide thin films are transformed into crystalline films. The authors use well-known characterization techniques like X-ray diffraction, scanning electron microscopy, ellipsometry, and UV-VIS spectroscopy to characterize the deposited films. The authors claim that the deposited films show good selectivity among other common reducing gases, like ammonia and carbon monoxide. The sensitivity has been proven to be highly dependent on the thickness and crystallinity of the samples.

The overall manuscript needs some minor revision before being accepted for publication. The following points have to be addressed by the authors.

·                 Fig.1 shows the XRD patterns of the e-beam WO3 thin films over fused silica substrates. Miller indices must be insert the on each peak shown in the figure to be sure that the obtained films agree with the reported results.

·                    as shown in Figure 3. The optical absorption spectra of nanogold-WO3 measured in air (black line) and during exposure to H2 (red line) Part (d) is very small (almost negligible) at 250 208 °C. How do the authors claim that the deposited Nanogold-WO3 can sense hydrogen?

·                    The authors show that the best performance shown in Figure 5d. This faster response is obtained at thickness of the nanoplates between 50-60 nm. This condition needs more physical explanation.  

Author Response

(The authors gave the same response as above.)

Round 2

Reviewer 2 Report

The response has answered the questions I raised. The revised manuscript is suitable for publication in its current form.